# Multicomponent Physical Exercise Training in Multimorbid and Palliative Oldest Adults

**DOI:** 10.3390/ijerph18178896

**Published:** 2021-08-24

**Authors:** Cristina Blasco-Lafarga, Gema Sanchis-Soler, Pere Llorens

**Affiliations:** 1Sport Performance & Physical Fitness Research Group (UIRFIDE), Physical Education and Sports Department, University of Valencia, 46010 Valencia, Spain; 2Department of General Didactics and Specific Didactics, University of Alicante, 03690 Alicante, Spain; 3Emergency Department, ISABIAL, General University Hospital of Alicante, 03010 Alicante, Spain; llorens_ped@gva.es; 4Department of Clinical Medicine, Miguel Hernández University, 03550 Sant Joan d’Alacant, Spain

**Keywords:** exercise intolerance, health care, home-based, hospitalization, walking aids, physical fitness

## Abstract

Exercise counteracts aging and pathology symptoms, but there is still scarce research on exercise programs for multimorbid and/or palliative old patients (MPO-Ps). In order to analyze whether the multicomponent physical–cognitive training is beneficial for this population, 17 MPO-Ps (81.59 ± 5.63 years) completed a >26 weeks home-based intervention (20–50 min/session, three sessions/week). Twenty-eight supervised and thirty-two autonomous sessions were gradually distributed along three phases: supervised training (ST), reduced supervision training (RST), and autonomous training (AT). Physical function (gait speed, hand grip and lower-limb strength, balance, and agility), mental status (MMSE), and autonomy in daily living (the Barthel Index) were assessed. Categorical analyses regarding the changes in the walking aids used in the test were added to improve the assessment of strength and agility along the intervention. Despite important study limitations, such as the small sample size and lack of a control group, and despite the MPO-Ps’ very low baseline fitness and initial exercise intolerance, they benefited from the dual-tasking approach, especially in autonomy, lower-limb strength, and balance. Agility improvements were shown only by categorical analyses. As expected, most benefits increased the supervision (ST phase). Gait speed and cognitive status maintained despite the total autonomy in training in the last phase. Further research with larger samples should confirm if multicomponent physical–cognitive exercise, individualized and tailored on daily-basis, together with technical assistance and medical supervision, benefits this MPO-Ps population, and if it can be prescribed to them with security, in spite some of them already being palliative patients.

## 1. Introduction

Sedentarism and physical inactivity are among the main risk factors susceptible of being modified to counteract the development of non-communicable diseases [1,2]. Illnesses such as obesity, diabetes, high blood pressure, cognitive impairment, or problems related to the musculoskeletal system do not depend on infectious agents, and are associated with the so-called inactive phenotype [3], pathological in the strict sense [4]. Besides, sedentarism and physical inactivity share direct and negative involvement in the aging process [1,4,5,6,7,8,9], enhancing major immobility, muscular power/mass loss [10,11], and the increase in inflammatory processes [4,12,13]. Inevitably, both are also associated with greater risk of falls [1,10], weakness increasement [10,11,14], the risk of becoming dependent or being admitted to a hospital [10,15,16], and, in the worst-case scenario, increased mortality risk between 20–30% [17]. This pathway of polypharmacy and hospital frequentation is particularly harmful to older individuals, triggering frailty, dependence, comorbidity, and a greater risk of mortality [18]. In addition, these multimorbid patients frequently experience pain, fatigue, dyspnea, depress mood, and anxiety in this last stage of lifespan, requiring palliative care [19]. Medication together with palliative care specialists, who are skilled at the management of limiting symptoms such as pain, are of paramount importance; however, palliative caregivers are still scarce, and both factors together increase sanitary burden [19].

Conversely, overwhelming evidence confirms the effectiveness of physical activity to counteract sedentarism, enhancing overall physical health, preventing and improving the medical condition in general, and the aging process in particular [5,20,21,22]. Exercise does not mitigate the aging process, but it attenuates many of its deleterious systemic and cellular effects, slowing many mechanisms involved in aging [5].

Besides the widespread consensus about the need to accumulate at least 30 min of moderate-intensity physical activity most days of the week [23], the World Health Organization (WHO) recommends both healthy and ill older adults to take part in moderate intensity multicomponent physical activities, or more vigorous whenever possible, three or more days a week [24], with special attention given to the effectiveness of the multicomponent training [24,25]. Recent studies also give light to promising outcomes of the benefits of exercise in very old ill patients despite their limited ability to tolerate exercise [12]. Regular exercise, especially strength training, benefits their metabolic system, for instance, lowering the glycosylated hemoglobin [11]. Their functional and cognitive systems improve, and they reduce their risk of falling and their fear of fall, resulting in better quality of life [4,11]. Concurrently, less sitting is as important as exercising [20,26], so these older adults need to keep active lifestyles and reduce the sedentary periods, especially in the afternoon [27]. This includes the time associated with an acute illness or a long stay in bed following adverse events [28].

Notwithstanding, current exercise recommendations [24,29,30] present some complexity in training programs addressed to very unfit ill seniors [12,23,25,31]. They require adjusting exercise to their individual characteristics and environment [11,12,25,31,32], their own or any of their siblings’ houses, long or short-term hospital units, or even the nursing homes. They may also need a different assessment approach, attending to those qualitative or small changes which are difficult to quantify or analyze with current tests and quantitative normal references. On the other hand, exercise interventions are scarce among chronic multimorbid and/or palliative old patients (MPO-Ps) (MPO-Ps) individuals due to their heterogeneity and psychosocial and family features. Furthermore, little is known about the viability and the positive impact of the physical–cognitive multicomponent programs in highly vulnerable seniors, with a high level of dependency. These multicomponent programs aim specifically to improve both physical and cognitive function, and might be of great interest due to this mixed nature [33,34]. Noteworthy, palliative older adults live all together with high rates of pain [19,23] and intolerance to exercise [35], hindering their exercise training.

Whether these adapted multicomponent exercise programs induce enough physical demands [11,12,23] and/or cognitive requirements to get positive adaptations in MPO-Ps, or whether this population requires continuous technical supervision to get any benefit, remains unknown despite its great interest. These thus become the main questions of this research.

More specifically, the present intervention aims to analyze the impact of a long-term (six months) home-based multicomponent physical–cognitive training program (MCCogTP) adapted for MPO-Ps; a particular sample of multimorbid, very deconditioned, and exercise-intolerant sedentary older adults, followed up at home by their doctors. Changes in physical function (i.e., gait speed, balance, upper and lower limb strength, and agility), as well as cognitive function and daily living autonomy, were considered. Categorical data analysis regarding the changes in the walking aids used in some tests were added to further understand changes in lower-limb strength and agility, following previous interventions [36]. This was due to the limitation of the “time” outcomes in some tests, such as the 30-s chair stand test (CST30; lower limb strength) and the timed Up & Go test (TUG, agility test). Physical improvements reduce the use of walking aids in these tests but leads to longer durations [29]. Finally, the effect of increasing autonomy/decreasing exercise supervision throughout the whole intervention was also considered.

As a main hypothesis, home-based adapted MCCogTP will maintain or even improve these three components of the overall health (physical function, cognition, and autonomy), at least during supervised training. Categorical analysis may give light to qualitative subtle changes which may be important in this stage of life.

## 2. Materials and Methods

### 2.1. Participants

Sixty-seven MPO-Ps from the Home Hospitalization Unit (HHU) of the General University Hospital of Alicante, were referred for admission to this pilot intervention. Once the doctor gave his approval and proposed the patients for inclusion, all patients and legal representatives were informed and voluntarily consented to participate. When any MPO-P was not able to sign/consent to participate, the relative/legal representative provided this consent. The study was approved by the Ethics Committee of the University of Valencia (H14014428868708), according to the Declaration of Helsinki.

As inclusion criteria: over 65 years of age, admitted or discharged from the HHU, with availability for follow-up after medical approval to participate. As exclusion criteria, those who refused to participate could not follow the training program for serious cognitive problems or followed another physical or rehabilitation program.

After the screening processes, 33 MPO-Ps started the study. Regrettably, the high experimental mortality (including voluntary abandon, hospital admission or pathology symptoms exacerbation, irregularity or non-compliance with the sessions, and death) reduced the final sample to those 17 who completed the 28 sessions of the MC^Cog^TP (81.59 ± 5.63 years; 9 women and 8 men) (Figure 1).

### 2.2. Experimental Procedure

After the initial screening, only when the MPO-Ps’ fitness and cognitive baseline outcomes were already analyzed, three mesocycles were designed following the guidelines of EFAM-UV© [33,37]. As previously described [33], EFAM-UV© psychomotor taxonomy sets six domains in the older adults’ functional retraining. Any task in a first level aims to improve at least one of two basic domains: postural control and gait, combined or not with manipulative and/or cognitive demands, which are complementary skills to reinforce in the first mesocycles. When the technical improvements allow to move on with security to the second level, EFAM-UV© introduces, little by little, exercise proposals on two more complex domains: rhythm and functional motor skills. Motor control, muscular resistance, cardiovascular demands, and executive function are thus higher in this second level, at the end of the macrocycle. Whatever the level, postural control and gait are the main objectives. On the other hand, the exercise intervention, followed the multicomponent and dual-task approach, setting individual progression by means of physical conditioning maps based the participants capabilities, as usually in EFAM-UV© [31,32,33,38,39].

Noteworthy, dual-task can be more useful and with a greater transfer to the life of older adults, thus being more effective than working on a single task [40].

Figure 2 describes the main contents of the mesocycles in this study. Training cycles and exercises were individualized and tailored accounting its three phases. All the participants achieved 28 sessions of supervised exercise and were provided with another 32 individualized and tailored sessions to do on their own.

Hence, the distribution of the sessions, which were always designed by a sport science graduate, sought a progression towards autonomy in exercise. The program started with two supervised, sessions plus one of autonomous exercise weekly (supervised training: ST) which then changed to one supervised and two autonomous sessions in the reduced supervision training phase (RST); and ended with three autonomous sessions in the third phase (autonomous training, AT), prevailing in this case of the telephone follow-up. As above mentioned, training objectives and programming were common, but sessions and exercises were daily individualized and tailored to each patient. Additionally, between the first two mesocycles, transition microcycles “T” were introduced for those patients who did not reach the objectives in the previous one (Figure 2).

Four testing periods (from EV_1_ at baseline, to EV_4_ at the end on the intervention) were set to analyze the autonomy-related changes in functional capacity, cognitive function, daily living autonomy, and health control variables along the intervention. The testing sessions were distributed in two non-consecutive days to avoid fatigue and testing interference, with an equitable distribution of questionnaires and functional tests, and at least one full week of training (i.e., three non-consecutive sessions) before any assessment. The whole intervention often took 26 weeks, including its 4 evaluations.

### 2.3. Functional Capacity Assessment

Patients were instructed to walk 4.5 m in a brisk but safe pace for gait speed analysis [14]. Walking aids were allowed (person, walker, or walking stick) when needed. A lack of space in some houses led to the removal of acceleration and deceleration zones.

Agility, static, and dynamic balance were assessed by the 8-foot timed Up-and-Go test (TUG), the Berg balance test, and the Tinetti test (this latter informative of gait, balance, and total Tinetti scores: _Ti_G, _Ti_B and _T_Ti, respectively). According to Rikli and Jones [41,42], patients walked 2.44 m in the TUG test, registering the time taken to complete it. Given the high state of frailty, only one attempt was made. If there was any technical or confusion error, an additional attempt was allowed. The Tinetti test (16 items; 7 for gait and 9 for balance) [43] and the Berg test (14 items whit a total score of 56 items) [44] allowed to determine the static balance and the quality of the gait.

Lower and upper body strength were evaluated with the 30-s chair stand test (30 s-CST) [41,42], and the hand-grip test (HG, dynamometer model T.K.K. 5401 Grip-D). Patients could use aids and/or modify the technique when required in the 30 s-CST. HG was determined in a 5 s maximum isometric contraction, two alternate attempts per hand with a recovery of 30 s. The best attempt was registered [45]. To ensure accuracy, the gait speed, TUG, Tinetti, and the 30 s-CST were recorded for video analysis by two observers (Kinovea-0.8.15 software).

### 2.4. Categorical Analysis of the 8-Foot Timed Up-and-Go Test and the 30-s Chair Stand Test

Some MPO-Ps were close to disability, so they needed walking aids and/or personal assistance to perform the 30 s-CST and the TUG test. Following previous research of the group [36], both tests were categorized accounting the walking aids used in each assessment. Four categories were determined for the TUG test analysis (unable; assisted by a person; assisted by an instrument; able without any help); and other five categories were set for the 30 s-CST (unable; able but incomplete movements and some aids; able without aids but still incomplete movements, complete movements with aids; able). Forward and backward changes of category among assessments (positive differences, ties, and negative differences) were therefore considered qualitative improvements or impairments on the functional capacity, respectively.

### 2.5. Cognitive Function, Daily Living Autonomy and Health Variables Assessment

Independence in activities of daily living and mental status were assessed by means of the Barthel Index BI [46] and the Mini-Mental State Exam (MMSE) [47], respectively; the first of them with 22 items and a maximum score of 88, and the second with 6 dimensions and a total score of 30.

Blood pressure (OMRON M3 model (IM-HEM-7131-E) Omron Healthcare Co., Ltd. Binh Duong, Vietnam), arterial oxygen saturation (SaO_2_ %, pulsioxímetro WristOx 2, Model 3150 pulse oximeter. Nonin Medical, Inc., Amsterdam, The Netherlands), and blood glucose (glucose monitor Accu-Chek Aviva. Roche diabetes care Spain, S.L., Sant Cugat del Vallès, Barcelona), together with height and body weight in those patients who could tolerate bioimpedance (Tanita BC545N. TANITA Corporation, Amsterdam, The Netherlands), were also retained as health control variables.

### 2.6. Intervention

Every multi-component workout (balance, strength, coordination and mobility, most times combined with cognitive demands, i.e., dual tasking) aimed to last over 50 min. When MPO-Ps were tired, felt pain and/or demotivation, effective motor-time was reduced to 20–30 min, and exercise was interspersed with recovery complementary tasks like playing cards, memory games, riddles or drawing activities, etc., until the 50 min. Elastic bands, small weights (both adapted to each patient), instability cushions, ropes, balls, training marks or floor lines and own-made materials were used for gait and postural control training [33,37]. Memory, association, inhibition, or decision-making tasks (i.e., executive function) enriched these two MC^Cog^TP basic motor domains [33,37]. Specific manipulative/handling skills, tailored for each patient, completed this adaptation of the EFAM-UV© training routines. Rhythm proposals were introduced at the end of the intervention (mesocycles E & D, Figure 2). All of them looked for joy and motivation. Asymmetry was prevalent for the functional improvement, both in strength and motor-control tasks.

Exercises, volume, and intensity were conditioned to the patients’ capacities, classified into high, medium, or low level, according to their dependence status, health, technical capacities, and movement velocity. According to the EFAM-UV© basis [32,33,37], there were one to three different exercises for the same motor target, so the progression in volume and intensity was guided by changes in the difficulty and complexity of the task. For instance, changes in the balance demands (sitting to sitting-tandem; sitting to standing; standing to tandem position, etc.), the level of coordination and cognitive demands, or changes in the range of movement and/or the movement velocity (slow and then quicker; short-isometric and then large and dynamic muscular contractions), among other constraints, were used to ensure variability in practice. In fact, EFAM-UV© avoided the exercise repetitions to increase the amount and type of stimuli in one session [32,33].

The sessions were divided into three sections, and both, the pre-post workout and the end of each section, were controlled by means of the modern Borg rating of perceived effort scale (RPE) [48], the EVA scale (visual scale for perceived pain), and the SaO_2_ (3 to 6–7 of 10 RPE, avoiding uncomfortable/painful situations. Controlling that SaO_2_ meant it did not fall below 95%. Blood pressure was also controlled in all patients, and at the end of each block in hypertensive participants. Practice stopped when hypertensive/hypotensive values were reached, or if there was an absence of adaptive changes during exercise [31,32].

### 2.7. Data Analysis

Considering physical exercise (i.e., MC^Cog^TP) as the treatment or main factor in this pre-post study, Gait Speed, HG, repetitions in the 30 s-CST, Berg, Tinetti, Barthel Index and MMSE were selected as quantitative-scalar dependent variables; whilst categorized 30 s-CST and TUG test were selected as qualitative-categorical variables.

Data were analyzed with SPSS v.22. (IBM, EE. UU). According to the Shapiro-–Wilk test, a repeated measures ANOVA was conducted for HG, TUG, and MMSE, followed by the Bonferroni post-hoc. The Friedman’s test was applied for 4.5-m gait speed, 30 s-CST, the Tinetti test, Berg and BI, followed by Wilcoxon paired comparisons.

Given that some individuals progressed in the aids to perform the tests (from greater to less dependence), but impaired their times to complete them, a categorical analysis was added for the agility (TUG) and the lower limb strength assessment. TUGs and 30 s-CST categorical analysis regarding the aids was performed with the non-parametric sign test.

Following Fritz, et al. [49], the Cohen d effect size, included for any significant change, was considered small (0.2), medium (0.5), and large (0.8). The *p*-value was established at *p* < 0.05, and trends (*p* < 0.1) towards significance were included [50].

## 3. Results

Baseline data confirmed advanced age (more than 70% were >80 years old), obesity, low SaO_2_, and low functional capacity in the MPO-Ps, together with the inability of some of them to carry out the tests (Table 1). The main diseases include cardiovascular, metabolic, respiratory, musculoskeletal, and cognitive illness (e.g., hypertension, COPD, asthma, osteoporosis, osteoarthritis, cognitive impairment, and cancer). These degree of disease and low physical fitness at baseline affected exercise proposals, intensities (which were lower than expected), and recovery times within exercises. Notwithstanding, the adapted MC^Cog^TP was successful and safe.

As shown in Table 2 (upper section), the Friedman test was significant for Berg (*p* = 0.05) and showed a trend towards significance for 30 s-CST (*p* = 0.07) and BI (*p* = 0.06). Further Wilcoxon paired comparisons showed a significant improvement and a medium effect size for Berg (39.68%; *p* = 0.038; *d* = 0.40), _T_Ti (34.30%; *p* = 0.046; *d =* 0.37), BI (31.44%; *p* = 0.037; *d =* 0.31), and 30 s-CST tests (78.18%; *p* = 0.023; *d =* 0.44) after ST (EV_2_). Changes in _Ti_B and _Ti_G were reduced to a trend (_Ti_B: 39.94%; *p* = 0.073; *d* = 0.38 and _Ti_G: 30.82%; *p* = 0.064; *d* = 0.33). Again, compared to baseline, the improvement and medium effect size persisted despite reducing supervision (RST; EV_3_), now for 30 s-CST (56.96%; *p* = 0.047; *d =* 0.31) and BI (33.35%; *p* = 0.045; *d =* 0.36). Conversely, the univariate approximation of the repeated measures ANOVA (Table 2, lower section) showed no significance for HG (HG_right_ & HG_left_), neither for MMSE or the quantitative measurement of the TUG test (Time in seconds).

The categorical approach (Figure 3) showed significant improvement in TUG after ST (EV_1_ vs. EV_2_; *p* = 0.016), followed by a trend once after RST (EV_1_ vs. EV_3_; *p* = 0.06). 30 s-CST displayed a trend (EV_1_ vs. EV_2_; *p* = 0.07) regarding the quality of its improvement after ST.

## 4. Discussion

Despite important study limitations, such as the small sample size and lack of a control group, MC^Cog^TP benefited the sample of MPO-Ps notwithstanding their very low functional capacity and level of dependency at baseline. The dual task approach might have helped to overcome their initial rejection and intolerance to exercise, engaging them along the 26 weeks of the intervention. As it was expected, supervised training provided the best improvements, and MPO-Ps improved in lower limb strength, agility, quality of gait and static and dynamic balance, although the effect size was moderate to small. Equally, the MC^Cog^TP significantly improved their autonomy in daily activities, although gait speed and cognition improvements were not significant. As a main finding, even though lower limb strength and autonomy were retained along the intervention, total autonomy seems inadvisable under this multicomponent physical–cognitive approach. As a second outcome, suspecting that agility and lower limb strength testing could be influenced by the qualitative progression in the task, more permissive categories of execution were stablished for 30 s-CST and TUG following previous researches [51,52], giving light to improvements in agility after ST hidden by the quantitative analysis.

The baseline assessments, as well as the inability of some MPO-Ps to carry out some of the tests, confirmed their very low physical fitness and functional capacity [14,41,42,53,54,55]. Gait speed was far below frailty limits [14], predicted survival [56], and incident disability [57], confirming the utility of this intervention in the setting of more reference values to describe the MPO-Ps population. Despite this very low starting point and the reduced exercise intensity below the initial prediction, this is the first long-term (>26 weeks) and home-based MCCogTP adapted for chronically ill, multimorbid patients, with some of them being palliative. Hereby, our data confirm a previous review by Cadore, et al. [58], pointing out the suitability and benefits of the multicomponent training in this population, at least under supervision. This is mainly because of the improvements on may hallmarks of frailty, even when these older people may have mild cognitive difficulties, as indicated by the MMSE results.

On the other hand, the combination of dual tasking, decision-making, and challenging constraints might have contributed to reduce the exercise intensity and even the volume of motor practice in the sessions, lowering its physical and physiological impact below the current recommendations [10,11,23]. Conversely, it increased the use of cognitive resources and joy, what might have helped to improve autonomy in daily activities, even along the RST phase. Indeed, MPO-Ps neither enhanced their cognitive function, nor reduced it after >26 weeks, which may represent a success in this population. In addition, together with the medical counselling, pharmacological treatment, and the multimodal approach of the intervention, the MC^Cog^TP may have been also helpful to increase adherence; an important challenge in this population [12,59]. It is also worthy to note the association between the improvements in physical function and autonomy in MPO-Ps and the significant decrease in their caregivers’ burden previously shown [32].

Going further in the lack of some meaningful changes, i.e., the MMSE or the gait speed, both of them related to the cognitive function [60]; this is more difficult to improve and requires longer training periods as compared to physical function in seniors [33]. It may also require future studies on the continuous supervision along any exercise intervention. Bring forward the starting of these multimodal strategies, including early exercise multicomponent physical–cognitive programs in the HHU, would be also advisable to ensure improvements in these capacities [31,32]. In addition, our MPO-Ps stopped the training sessions whenever they went to the hospital or displayed some acute increase in their pathology symptoms. Adverse events always resulted in a training/fitness setback. The larger the stop and/or the lower the functional and physical level of the patient, the larger the setback.

As already mentioned, the decrease in the supervision process might be co-responsible of some plateau effect along the intervention. Geriatric fitness trainers minimized the exercises’ demands and ambulation requirements to maximize safety in the autonomy sessions. Moreover, given that most times training spaces in home-based interventions were tight, lack of proper illumination or with obstacles, which affects the sensory stimuli and the neural drive, exercise proposals were simplified in the AT phase. Therefore, the maintenance of certain capacities and/or the absence of significant losses might be considered successful in this scenery [48], though continuous supervision seems advisable.

On the other hand, regarding the second, but not less important, aim of this study, the qualitative approach in the categorical analyses shed light on improvements in strength and agility hidden by numbers. This confirms its relevance in the assessment and understanding of slight changes which may be related to better independence and perceived autonomy [12] in the long term.

To conclude, there are some limitations in our study, mainly the small final sample size and the lack of control group. Furthermore, there are various phenomena related to the poor health conditions of the MPO-Ps, the interruption in the continuity of the sessions due to adverse events, or their large heterogeneity regarding the needs and type of exercise. Moreover, home-based interventions, including exercise, are not so usual in the HHU, what have demanded extra work and continuous adjustments along two years.

The fact that it was a pilot study together with the challenge of replicating a similar sample, and lack of financial resources, blunted the possibility of the control group. On top of that, doctors opted to include patients with an existing moderate cognitive impairment with the intention of including the maximum number of patients likely to benefit from the exercise, which, in general, might reduce the quantitative benefits of our MC^Cog^TP. Establishing minimum MMSE > 20 range, lowering the minimum age to be involved in these home base programs, or including senior adults with a faster gait speed, for example, would improve the expectations of improved quality of life, although it would denature the aim of the researchers and the palliative care medical service, focused on uplifting and preserving the quality of life of these MPO-Ps until the end of their lives as much as possible.

## 5. Conclusions

On the one hand, the dual-tasking approach in the MCCogTP was not an inconvenient for the improvement of physical function and daily living autonomy in MPO-Ps derived from the HHU. The absence of significant impairments in gait speed or cognitive function after six months might be also a positive outcome in this population. Constraints, challenges, and joy might have helped to get over 26 weeks of adherence to exercise in this MCCogTP; despite this, both technical supervision and daily exercise tailoring are required to ensure some of these improvements. On the other hand, the categorical data analysis regarding the walking aids in the test improves the quality of the functional assessment in this frail population.

## Figures and Tables

**Figure 1 ijerph-18-08896-f001:**
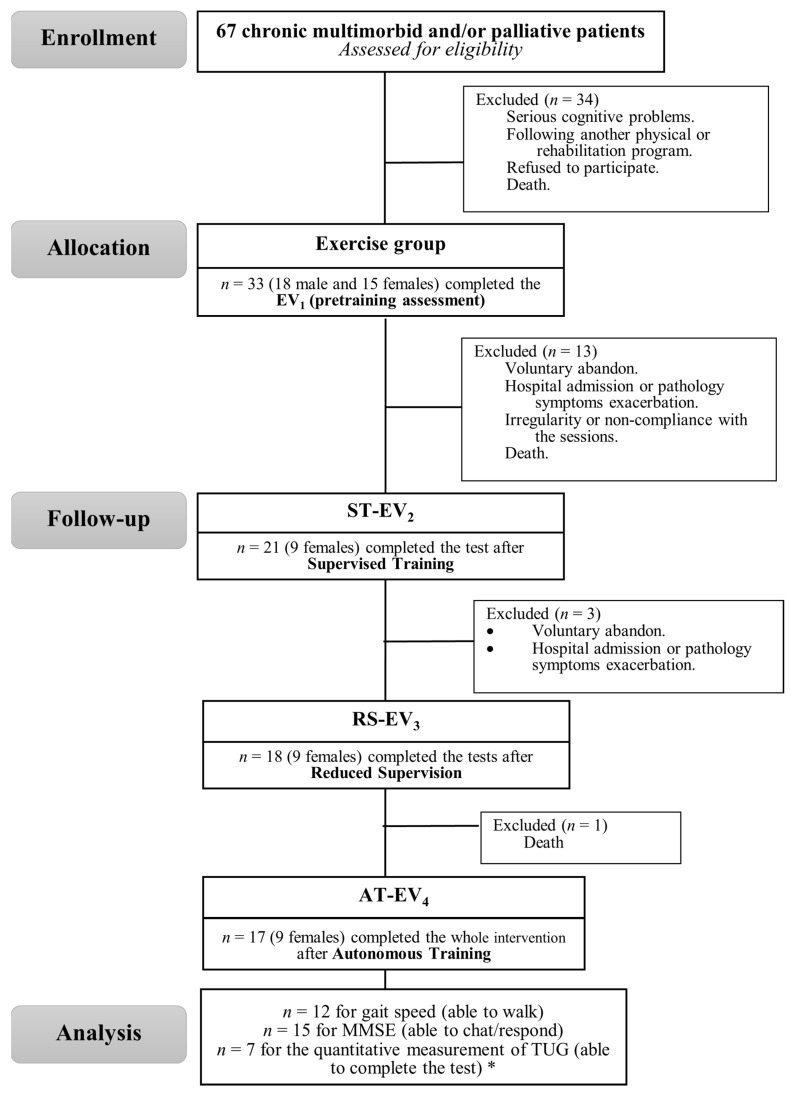
Flow chart of participants. ST: supervised training; RST: reduced supervised training; AT: autonomous training. MMSE: Mini Mental State Exam; TUG: 8-foot Timed-Up & GO test. * With or without walking aids.

**Figure 2 ijerph-18-08896-f002:**
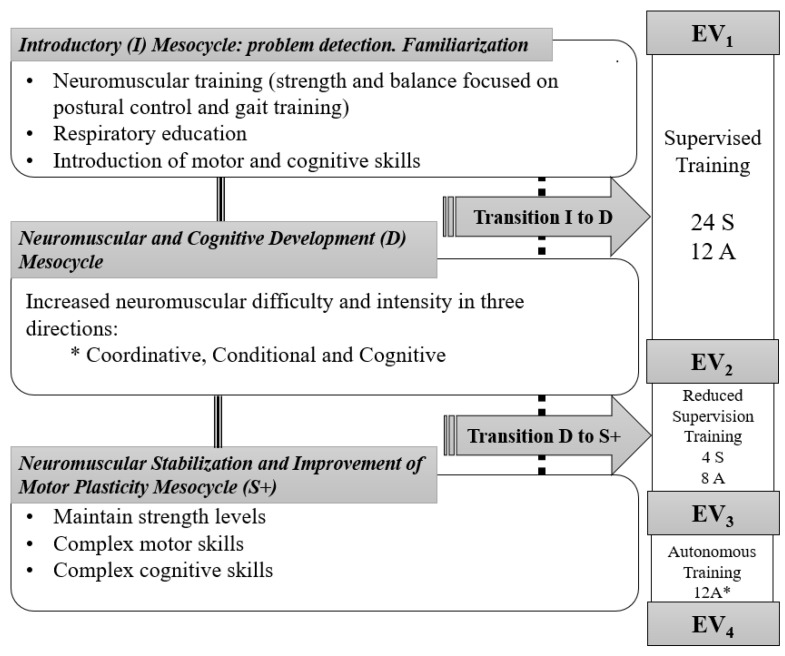
Phases and planning according to mesocycles of the training period. * During the autonomous phase. Telephone follow-up; uncertainty about the 3-weekly sessions completion in some subjects. Black line: training process; Dash line: training process when transitions were needed; S: Supervised sessions; A: Autonomous sessions.

**Figure 3 ijerph-18-08896-f003:**
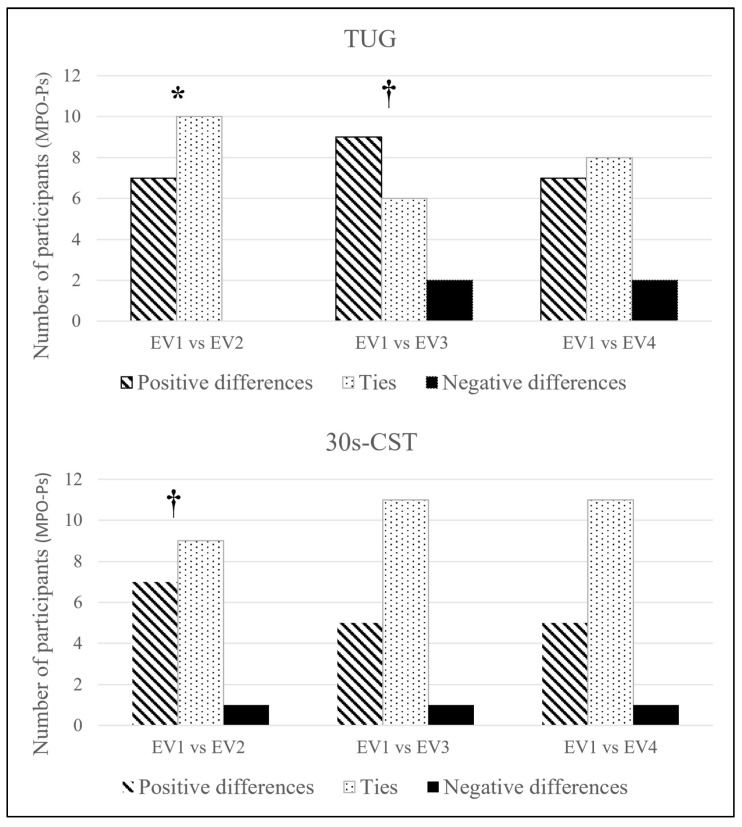
Categorical analysis of the TUG and 30 s-CST. Sign test. Abbreviations: 30 s-CST: Thirty seconds Chair-Stand test; TUG: 8-foot Timed-Up & GO test. * *p* < 0.05; ^†^ *p* < 0.1.

**Table 1 ijerph-18-08896-t001:** Descriptives (*n* = 17): mean, standard deviation, coefficient of variation, and frequencies.

	**Mn ± SD**	**CV%**
Age	81.6 ± 5.6	6.9%
SBP (mmHg)	133.0 ± 15.07	11.3%
DBP (mmHg)	71.5 ± 9.5	13.3%
Weight (Kg) ^a^	69.5 ± 13.8	19.9%
Height (cm) ^b^	154.1 ± 12.6	8.2%
BMI	30.1 ± 5.2	17.3%
SaO_2_ (%)	92.7 ± 5.3	5.68%
	**Frequency**	**Percentage**
Gender		
Male	8	47.1%
Female	9	52.9%
Walk initially		
Yes	12	70.6%
No	5	29.4%
Pathological condition		
Chronic-Pluripathological	15	88.2%
Palliative-Oncological	2	11.8%

SBP: systolic blood pressure; DBP: Diastolic blood pressure; BMI: body mass index; SaO_2_: oxygen saturation. ^a^
*n* = 14 (3 subjects did not undergo this measurement because of pacemakers, stents, or metal plates or plates). ^b^
*n* = 15 (2 subjects were unable for standing upright without support).

**Table 2 ijerph-18-08896-t002:** Quantitative changes (mean and standard deviation) in physical and mental status: The Friedman and Wilcoxon test (upper section); ANOVA repeated measures test (lower section).

Test	N	EV_1_	EV_2_	EV_3_	EV_4_	*p*
Gait speed (m/s)	12	0.33 (0.22)	0.34 (0.22)	0.31 (0.25)	0.33 (0.25)	0.42
30 s-CST (Rep)	17	1.65 (2.87)	2.94 (2.91) *	2.59 (3.10) *	2.56 (2.86)	0.07 ^†^
Berg (s)	17	16.00 (14.25)	22.35 (17.00) *	21.18 (15.24)	18.00 (13.91)	0.05 *
Tinetti Balance (s)	17	5.71 (5.23)	7.47 (5.41) ^†^	6.76 (4.53)	6.41 (4.65)	0.24
Tinetti Gait (s)	17	3.53 (3.50)	4.94 (3.73) ^†^	4.65 (3.97)	4.47 (3.76)	0.45
Total Tinetti (s)	17	9.24 (8.10)	12.41 (8.87) *	11.41 (8.23)	10.88 (8.28)	0.13
Barthel Index (s)	17	30.88 (29.85)	40.59 (31.61) *	41.18 (26.84) *	38.82 (29.29)	0.06 ^†^
						***p***
Hand Grip Right (kg)	17	12.11 (6.63)	12.62 (6.31)	11.97 (6.70)	11.14 (6.29)	0.73
Hand Grip Left (kg)	17	10.43 (6.68)	11.04 (5.80)	9.78 (5.98)	9.88 (4.69)	0.34
MMSE (s)	15	24.60 (3.83)	25.27 (3.63)	24.93 (3.69)	25.20 (4.39)	0.93
TUG (s)	7	35.44 (26.45)	31.52 (12.78)	51.03 (38.79)	43.11 (28.59)	0.36

Abbreviations: _Ti_G; Tinetti gait, _Ti_B; Tinetti balance, _T_Ti; Total Tinetti, 30 s-CST; 30 s chair stand test, MMSE; Mini-Mental State Examination, TUG; 8-foot up-and-go, Kg; kilograms; Friedman: * significantly difference *p* < 0.05; ^†^ Trend to significance: *p* < 0.1; Wilcoxon: Significant improvements compare to baseline * significantly difference *p* < 0.05; ^†^ Trend to significance: *p* < 0.1.

## Data Availability

According to MDPI Research Data Policies, the data presented in this study are available on request from the corresponding author. The data are not publicly available due to privacy medical reasons.

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
