# Peer review of "Multicomponent Physical Exercise Training in Multimorbid and Palliative Oldest Adults"

_ijerph, 2021, doi:10.3390/ijerph18178896_

Round 1

Reviewer 1 Report

Many thanks for the possibility to evaluate the manuscript after revision. The manuscript and its readability have improved considerably. However, it would further benefit from a final language check from a native speaker – especially the introduction.

The research topic is very relevant and the research team has made considerable effort for the pilot study. Considering the nature of the study, limitations in the design and recruitment, some conclusions (especially in the abstract) appear not to be flowing from the data. Finally, please could the authors make sure consistency in using terminology e.g. multimorbid/very ill ect. Further detailed comments can be found below.

Abstract

Line 15: multimorbid and/or palliative old patients – this should be consistently referred as patients with…

Line 24:” and despite the MPO-Ps' very low baseline fitness and initial exercise intolerance” – Not limitation describes research population characteristics.

Line 30: “is safe and could be prescribed” – Please could you soften the conclusions considerably.

Line 41: “directive” – do you men direct?

Line 50: “…stage of lifespan becoming palliative.” Still difficult to understand what authors refer to with becoming palliative. Please add short explanation, i.e. requiring palliative care????

Line 51: “Medication together with palliative care providers, skilled at the management of these symptoms, particularly pain, are of paramount importance, but they are still scarce and increase sanitary burden.” Please rephrase – this sentence (or indeed word choice) does not make any sense.

Line 57: “…improving many mechanisms involved in aging.” Slowing?

Line 61: Numbers from 0-9 often written in text

Line 67: usual expression in this context is better quality of life – not healthier.

Line 68: “Concurrently…” please consider revising the sentence or breaking in more understandable, shorter, sentences.

Line 82: “…these programs…” – unclear to which programs are referred to. This sentence is overall very unclear.

Line 84: “Noteworthy…” – Very unclear what is the purpose of this sentence and if a reference is missing.

Line 89: “…becoming objectives…” revise to make clear that these are the study objectives.

Line 95: “Categorical data…” this and the next sentence do not read like research aims. Reader is also left to wonder about the mention of the previous study. Is this meant as a hypothesis??

Methods

Line 121: “experimental mortality” – is this a standard use of language? Or do the authors refer to participant drop-out / non-completion?

Results

Trends towards significance should not be included – they are insignificant results.

Discussion

356: “in the setting of new normal values to describe the MPO-Ps population” – this is confusing – do the authors mean that the values derived from this study should be used as new benchmarks? Would suggest considerably to tone the sentence down.

Reviewer 2 Report

Dear authors,

The manuscript is much better now. All the changes were worth it.

I hope I have contributed to some improvements.

Congratulations for the good work. 

Author Response

Dear reviewer

Thank you so much for your favorable review  of the manuscript ijerph- 1333993, entitled: “Multicomponent physical exercise training in multimorbid and palliative oldest adults”,

Once more we want to thank you for your contribution to the quality of the current manuscript. You can be sure that you have helped us to improved it.

The authors

This manuscript is a resubmission of an earlier submission. The following is a list of the peer review reports and author responses from that submission.

Round 1

Reviewer 1 Report

Many thanks for the opportunity for reading this manuscript. The manuscript examines effects of physical and cognitive training for fragile old adults. Considering the aging population, this is an important research area. The research design appears appropriate, but in its current form the manuscript is unfortunately in some places almost unintelligible to read. The use of language especially in introduction, discussion, and conclusion sections make the evaluation of the manuscript very difficult. There are also a number of places where information appears to be missing or use of language makes it challenging to evaluate what the authors mean. I have included some further comments below, but reserve the right to include more.

Authors would benefit from having a native speaker to read the manuscript through and check the use of correct terminology.

The title of the manuscript is very difficult to understand. Supervision and categorical data analysis in a sentence make very little sense. Also, the authors should consider their language use – for example “in multipathological palliative oldest” – what does this mean? Furthermore, diseases should not be used to describe individuals- this should be e.g. individuals aged 65 and older with multimorbid diseases

Abstract

It is unclear what do the authors mean with pathological symptoms – caused by diseases like cancer or rather cumulative effect of chronic diseases. Please clarify. Similarly, with palliative – are these individuals receiving palliative care? Aims should be clearer written. Conclusions do not stem from the results.

The abstract is overall very difficult to read and understand. Authors should consider comprehensive rewrite with English language checking.

Introduction

Line 35: Sorry, I am again a bit lost with the term pathological in this context. Please could the authors consider further explanations.

Line 38: both what? Sedendarism and inactivity?

Lines 42 – 44: improving what medical condition? Or do the authors refer to overall physical health? Also does physical activity counteract ageing?

Line 47: please define what is meant by seniors in this context.

Line 48: what is meant by special attention to multicomponent training? Who should pay attention? And Why?

Line 58: what recommendations? The research results above?

Lines 61 to 68: Are there any references missing?

Line 69: The sentence is not clear – is this a research aim?

Please could the authors add clear research question and hypothesis that were tested.

Also, please could the author refer from including methodological information, e.g. categorical data analysis, in here.

Methods

Was the participation voluntary? Using the word “by order” is very strong and indicates involuntary referral. It would be also beneficial to state inclusion and exclusion criteria at the beginning.

What is experimental mortality? Times low MMSE test results as stated by authors – could the participant give informed consent or did they require assistance? Similarly, did participants receive any help in filling the questionnaires?

Line 100: Please could you be more specific – the cycles designed after the participants were included? What are the guidelines mentioned – please could a brief explanation be added.

Line 105: This sentence is very difficult to understand – please redraft.

Line 115: Sampling condition? What is this.

Line 127: Does this refer to how function capacity was assessed- please consider changing the sub-heading to better reflect the information in the section.

How were the categories decided for the categorical analysis (post hoc?)? How many participants were in each of the categories. Were Bonferroni corrections used due to multiple testing, e.g. TUG? Why were trends towards significance included? Please could the authors include the rationale for this – especially if this was a post hoc decision. Considering the use of both parametric and non-parametric tests, did the authors examine data normality?

Also considering the importance of the authors laid on the categorical analysis, please could you include more information – including the hypothesis for this analysis in the introduction.

Please could the authors explain why participants are in the table reported as those 70-79 and over 80? Other variables reported are not stratified according to age.

Discussion

A number of new references is introduced in the discussion – largely, these references should already have been included in the discussion to aid the reader to understand the research question setting.

Line 255: Please consider redrafting – this end of the paragraph is very difficult to understand. I am also struggling to understand whether authors consider the categorical analysis as a qualitative analysis – this is partially due to lack of information in both methods and results sections.

Line 246: …confirming the need… - Was this one of the aims and how these results confirm the need for new values?

Line 294: Is there a reference missing? Or is this a result from this study?

I will stop the commenting here and refer to overall comments above.

Reviewer 2 Report

The work is interesting, well designed
some suggestions

you can make a diagram according to CONSORT on the study design. Thank you

Are the guidelines that you used validated in this type of population?

can improve the quality of figures

In figure 4, are the data average? is the standard deviation missing?

Reviewer 3 Report

Dear authors,

This study highlights the lack of studies among chronic multi-pathological and palliative individuals due to their heterogeneity and psychosocial and family features. Thus, the purpose of this investigation was to analyse the impact of increasing autonomy/decreasing exercise supervision throughout a home-based multicomponent physical-cognitive training program adapted for palliative individuals. The study is apparently well conducted.  

Abstract 

The abstract is clear. 

Introduction

L39: The authors showed references for both “risk of falls” and “the risk of becoming dependent or being admitted to a hospital”. However, the authors didn’t show any reference for “weakness increment”

Materials and Methods

L83: The authors should start the sentence with neither numbers nor abbreviations. I would like to suggest changing the sentence’s structure. 
2.3 Functional capacity 

L128: “Patients were instructed to walk 4.5 meters in the shortest time possible for gait speed analysis [30].” What do the authors mean by “shortest time possible”? Reference 30 (Fried et al. 2001) carried out this test by asking the participants to walk 15 feet at the “usual pace”. The authors must make clear what instructions they gave to the participants and how the test was carried out.

Figures 2, 3 and 4. 
Figures are always welcome given their ease in illustrating the details of the study. However, figure 3 is extremely hard to read the sentences provided by the authors. A better resolution is needed. Although figure 2 is better than figure 3 in terms of resolution, if it is possible, a better resolution should be given as well.  Figure 4 can also be improved. 

L:139 The authors used the 30s-chair stand test to assess participants’ lower body strength.  Recently, some authors have used 5 times chair stand test to assess lower body muscle strength. The purpose of this test is to measure the amount of time needed for a patient to rise 5 times from a seated position without using his or her arms. The 5 times chair stand test is a variation of the 30s-chair stand test. Since the original test requires both strength and endurance, the 5 times chair stand test has been proposed as convenient to measure strength. Thus, the authors should also mention that the 30s-chair stand test assesses endurance and not only strength. 

I noticed a lack of justification for working in a dual-task. The evidence has been suggesting that the dual-task can be more useful and with a greater transfer to the life of the elderly, thus being more effective than working on a single task. The authors could include a justification.

A suggestion of a reference: 
doi: 10.1016/j.apmr.2008.09.559

Results
The results are simple and easy for the readers. 

Discussion
The discussion is clearly presented and the conclusion is supported by results. 

L305: "Shed light to" I would suggest changing it to "shed light on". Please consider this comment just a suggestion

The methods used for defining the intensity should be discussed. I attach a suggestion that might help:

DOI:10.1080/14779072.2019.1561278

Reviewer 4 Report

Manuscript ID: ijerph-1240930

Supervision and categorical data analysis: required strategies to enhance training and testing in multipathological palliative oldest following a multicomponent exercise intervention

This pilot study has aimed to analyze the physical and cognitive impact of a home-based multicomponent physical-cognitive training program adapted for chronic multipathological and palliative older adults, especially focusing on the supervision and increasing autonomy during the training.

First of all, I would like to really congratulate the authors for this work, since the complexity of designing and implementing a training program for this kind of population is a really complex endeavor, it is so difficult to get adherence. Especially, the fact of individualizing the exercises and the whole intervention to diseased people. I have read previous studies of the authors and the training program is well designed and completely individualized to the pts.

Although the manuscript is well-written and design, I would like to point some points to be taken into account by the authors:

Firstly, the title does not express the real content of the study. It is not clear at all, one does not know what supervision would mean here and categorical data analysis is not a strategy to enhance the effects of the intervention, but just an analytical strategy to see some covered positive results, so a mediator in the relationship. Authors should consider changing it.

The quality of the images is very poor; a general improvement must be carried out.

ABSTRACT

- Line 20: It is not clear to me how many sessions the pts finally performed. Here, it is said more than 26, 28 in the Participants section (line 87), 28 + 32 in Experimental procedure (lines 102 and 103). You should state clearly how many total sessions were performed by each pt. Or maybe this minimum of 28 sessions to get the training completed was established? This aspect is not well explained, or at least I did not get it well. Please, clarify.

- Line 25: Which improvements the authors are referring to? You should specify them because from my point of view this is one of the main strengths of this work.

- Authors should include something regarding the main limitations of the study in the Abstract. Something like: “Despite the important limitations such as the small sample size, lack of a control group...... …the results showed………”

INTRODUCTION

In the first paragraph of the Introduction, the authors are talking about aspects highly related to the topic of frailty. In fact, they discuss some of their results under this frame in the Discussion. So, this entity should be named here, giving more strength to this study. Actually, along my reading, I was wondering why the authors did not assess frailty.

MATERIALS AND METHODS
- Participants: multipathology and palliative. This is a highly heterogeneous sample, it may be a mix of a lot of pathologies, and this could affect the results. Although the authors give a clue in Table 1, it would be a good idea to detail the profile of the participants, which kind of pathologies. Especially because the inclusion criteria are very general.

- Experimental procedure: the issue with the number of sessions must be detailed.

Sections 2.3, 2.4, and 2.5 are referring to the evaluations, and this is not clear, maybe the Authors should name section 2.3. Evaluation and set the current sections as sub-sections.  

Line 146: there is a typo, “8-food” must be replaced by “8-foot”

- Intervention: Although the intervention is deeply explained elsewhere, I consider that adding a couple of images with examples of the tasks would be a good idea.

- Data analysis: The dependent variables, factors, and levels of the statistical analyses must be indicated.

The type of effect size measure for each kind of statistic must be indicated.

RESULTS

I understand that a statistical result of 0.07 is so close to a significant result that is a pity not to consider it as a trend. However, from my point of view is a practice that should be abandoned, as regretfully nothing above 0.05 must be considered as significant. In fact, the effect sizes are also too small.

Table 2: What is K? You said Friedman in the foot table, but are you referring to some statistic or p-value? In the ANOVA section you refer to p, not to F so to homogenize please use the same logic.

DISCUSSION

Although the statistical results are modest, there are some benefits. But in general, the conclusions given in the Discussion are too robust given the present results. So, please be more cautious.

Which particular result from this study supports the conclusion that the dual-task approach was useful to overcome rejection and intolerance to exercise? No data regarding adherence or satisfaction level are analyzed here.

Authors conclude that an improvement in the basic daily activities state was found, but the results for Barthel were a trend, so that was no significant.  Although it is true some results were significant in the paired comparisons, the general Friedman test was not.

Lines 263 and 270: did you mean “frailty” when you say “fragility”?

Line 270: What you mean with low mini-mental levels? A low score in the MMSE, is this not the same to say cognitive impairment?

Lines 277-278: That is a good observation, but regrettably the period between evaluations is not enough to ensure this kind of statement.

Lines 284-285: I would say cognitive function in general, as MMSE is not just related to executive function as with other cognitive domains too.

Lines 304-308: This is a nice conclusion. Most of the research in this field just relies on quantitative approaches whereas the qualitative approach is of extreme importance especially when working with disabled samples.

Finally, the Authors acknowledged all the shortcomings along the study. So, some ideas I had to improve the paper whereas I was reading were already addressed here.